# Influence of atomic site-specific strain on catalytic activity of supported nanoparticles

Torben Nilsson Pingel [1,2], Mikkel Jørgensen [1,2], Andrew B. Yankovich[1],
Henrik Grönbeck [1,2] & Eva Olsson [1,2]

Heterogeneous catalysis is an enabling technology that utilises transition metal nanoparticles (NPs) supported on oxides to promote chemical reactions. Structural mismatch at the NP–support interface generates lattice strain that could affect catalytic properties. However, detailed knowledge about strain in supported NPs remains elusive. We experimentally measure the strain at interfaces, surfaces and defects in Pt NPs supported on alumina and ceria with atomic resolution using high-precision scanning transmission electron microscopy. The largest strains are observed at the interfaces and are predominantly compressive. Atomic models of Pt NPs with experimentally measured strain distributions are used for first-principles kinetic Monte Carlo simulations of the CO oxidation reaction. The presence of only a fraction of strained surface atoms is found to affect the turnover frequency. These results provide a quantitative understanding of the relationship between strain and catalytic function and demonstrate that strain engineering can potentially be used for catalyst design.

[1] Department of Physics, Chalmers University of Technology, 41296 Gothenburg, Sweden. [2] Competence Centre for Catalysis, Chalmers University of Technology, 41296 Gothenburg, Sweden. These author contributed equally: Torben Nilsson Pingel, Mikkel Jørgensen, Andrew B. Yankovich. Correspondence and requests for materials should be addressed to E.O. (email: eva.olsson@chalmers.se)

Metallic nanoparticles (NPs) supported on oxide substrates are critical material architectures for various applications, including catalytic synthesis of chemicals[1], catalytic abatement of air pollution[2] and catalysis for sustainable energy systems[3]. In order to enhance catalytic efficiency, reduce operating costs and meet the perpetually more stringent emission regulations, there are strong motivations to more completely understand and control the factors that influence catalytic activity of supported NPs. Presently, nanocatalysts are routinely tailored towards specific reactions and conditions by varying NP size, composition and support material[4–12].

It is well established that strain modifies the chemical properties of metallic systems[13,14]. However, the effect of NP strain on their catalytic activity has not been fully explored and offers potential for further leverage in designing catalyst systems. The strain in a NP can be intrinsic or extrinsic. Intrinsic strain is caused by the structure of the catalytic NP material itself, and arises, for example, because of the finite size[15,16], morphology[17] or domain structure[18–20]. Alternatively, strain can be caused by extrinsic factors such as inducing lattice mismatch at an interface within a core–shell structure[21–26] or at a NP–support interface[27–31]. Previously, strain in supported NPs has been characterised using X-ray diffraction-based techniques[27,28], where atomic site-specific information is inaccessible due to NP averaging and low spatial resolution. Atomic-resolution transmission electron microscopy studies have revealed local strain variations in NPs due to the support interface[29–31]. More detailed knowledge of site-specific strain variations requires better precision in locating atom positions and could open up additional avenues to better understand catalytic properties.

The influence of strain on reaction energies has been rationalised by the d-band model[32] and investigated in numerous theoretical studies on extended surfaces[31,33–37]. However, the resulting predictions from reaction kinetics studies are vastly different when comparing NP structures to extended surfaces[38], and the influence of strain on reaction kinetics over NPs has not been explored. It is therefore not clear how the impact of interfacial strain on catalytic activity compares to other factors such as chemical properties of interfacial sites.

Here, we demonstrate that the complex relationship between strain and catalytic activity of supported NPs can be uncovered by using precise strain measurements as inputs for first-principle-based kinetic simulations. We perform high-precision scanning transmission electron microscopy (STEM) imaging of supported Pt NPs to enable site-specific strain mapping at atomic resolution and <0.7% strain precision. Our results reveal moderate 1–3% strains associated with grain, surface and interface structures, and strong but localised 3–10% strains primarily at the NP–support interface. Kinetic Monte Carlo simulations of NPs that incorporate the experimentally measured strain distributions reveal that the inclusion of strain affects the light-off temperature and attainable activity. Combining precise strain measurements and kinetic simulations opens up new possibilities for understanding the cause and implications of strain in supported catalytic NPs and could provide an additional handle to tune catalytic properties.

## Results

**Local strain variations in supported Pt NPs.** Atomic resolution strain measurements were extracted from high-precision STEM images that were obtained by acquiring image series of tens to hundreds of consecutive high-angle annular dark-field (HAADF) STEM images of the same supported Pt NP, followed by non-rigid image registration to remove image distortions and image series averaging to improve the signal-to-noise ratio (SNR). This procedure follows previously developed methods[39–41] but uses considerably lower electron beam currents (~3 pA) and total doses ($5 - 13 \times 10^4 \, e^- \, \text{Å}^{-2}$) in order to minimise electron-beam-induced damage. The atomic column positions in the high-precision images were measured with sub-pixel accuracy using two-dimensional Gaussian fitting, resulting in image precisions between 1 and 4 pm. These positions were then later used to extract information about local displacements and strain.

High-precision side-view STEM images of Pt NPs supported on γ-alumina and ceria (Fig. 1a, c, e, g) reveal the NP–support interfaces (see Supplementary Note 1 for more details). Based on observations of hundreds of NPs, the ones shown in Fig. 1 are typical in size and morphology of the respective specimens. The atomic structure of the alumina support is not visible because it is not in a zone-axis orientation and shows weak atomic weight Z-contrast in HAADF STEM images compared to Pt, while the ceria support is clearly resolved. The experimentally observed NP structures are in accordance with previous observations and predictions of stable face-centred cubic NPs. The NPs have primarily {111} and {100} vacuum surface facets, which have been identified earlier as the predominant facets on twinned fcc particles[42]. Twin boundaries are observed in a majority of the investigated NPs and are predicted to be stable under typical reaction conditions because they add little energy compared to single-crystalline NPs[42–44]. There are concave re-entrant surfaces at the twin boundaries on some NPs (for example NP 2, marked by an arrow) that have been shown to reduce the overall energy of twinned NPs[42,43]. The NP supported on a {111} ceria facet (NP 4) has an asymmetric decahedral shape that can reduce the NP's total energy by balancing the twin boundary, surface and interface energies[44–46]. A {711} Pt facet that is contained in a single grain spans the entire NP–ceria interface. Periodic interface dislocations and a ~4.5° tilt of the {111} planes are present at this interface due to the ~28% lattice mismatch between ceria and Pt.

Atomic column displacement maps (Fig. 1b, d, f, h) visualise the lattice deformations across an entire NP by displaying the shifts of the experimental atomic column positions from where they would be in the absence of any lattice deformations. The displacement maps reveal similar trends across all of the measured NPs: the interior regions of each grain are relatively deformation free, whereas displacements are present close to the twin boundaries, the NP surfaces and the NP–support interfaces. The {311} twin boundary in NP 3 (marked by red rectangle) induces considerably larger displacements compared to the {111} twin boundaries (marked by blue rectangles) from any data set. Both types of twin boundaries are metastable with local energy minima, but the energy for the {311} boundary is higher, making it less stable[47].

Displacement maps are a good visualisation tool for the overall behaviour of the lattice deformations across an entire NP. However, the smaller local atomic-scale lattice deformations are mostly hidden by the larger global deformations. Local strain mapping is a more suitable visualisation tool for local lattice deformations because it does not rely on an extended perfect reference lattice, as discussed more in the Methods section.

Projected strain maps (Fig. 2a–l) and bond angle maps (Supplementary Note 2) visualise and quantify both intrinsic and extrinsic strain in the NPs. The NPs show intrinsic strain at the vacuum surfaces that can be compressive or expansive and is most pronounced at edge and corner sites[41]. There is also consistent 1–2% intrinsic expansive strain in the atomic planes neighbouring the {111} twin boundaries in the directions perpendicular to the boundaries, but not parallel to them. The {311} twin boundary (NP 3) induces large periodic strains at the boundary and in the surrounding regions. The extrinsic strains at the NP–support interfaces are considerably larger compared to

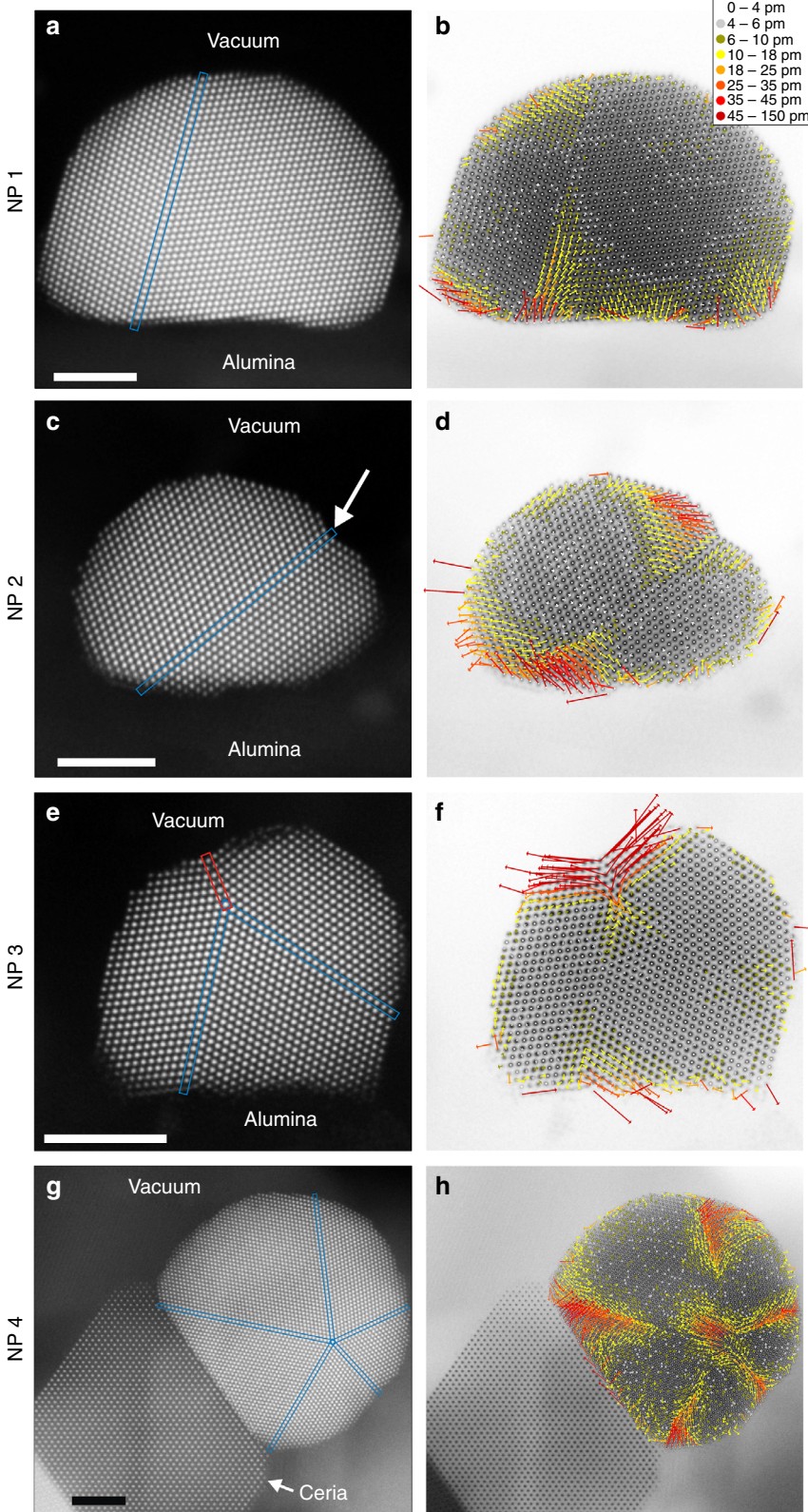

**Fig. 1** High-precision atomic-resolution STEM images and displacement maps of supported Pt NPs. Left column, **a**, **c**, **e** and **g**: : High-precision side-view HAADF STEM images of Pt NPs along ‹110› supported on alumina (**a**, **c**, **e**) and ceria (**g**). The interfaces between the NPs and the support are seen at the lower edge of the NPs. The arrow in **c** marks a concave re-entrant surface. Right column, **b**, **d**, **f** and **h**: Displacement maps showing the deviation of each atomic column position from their unstrained lattice positions. The arrows display the direction and magnitude of each displacement, as indicated in the legend. Each scale bar corresponds to 3 nm

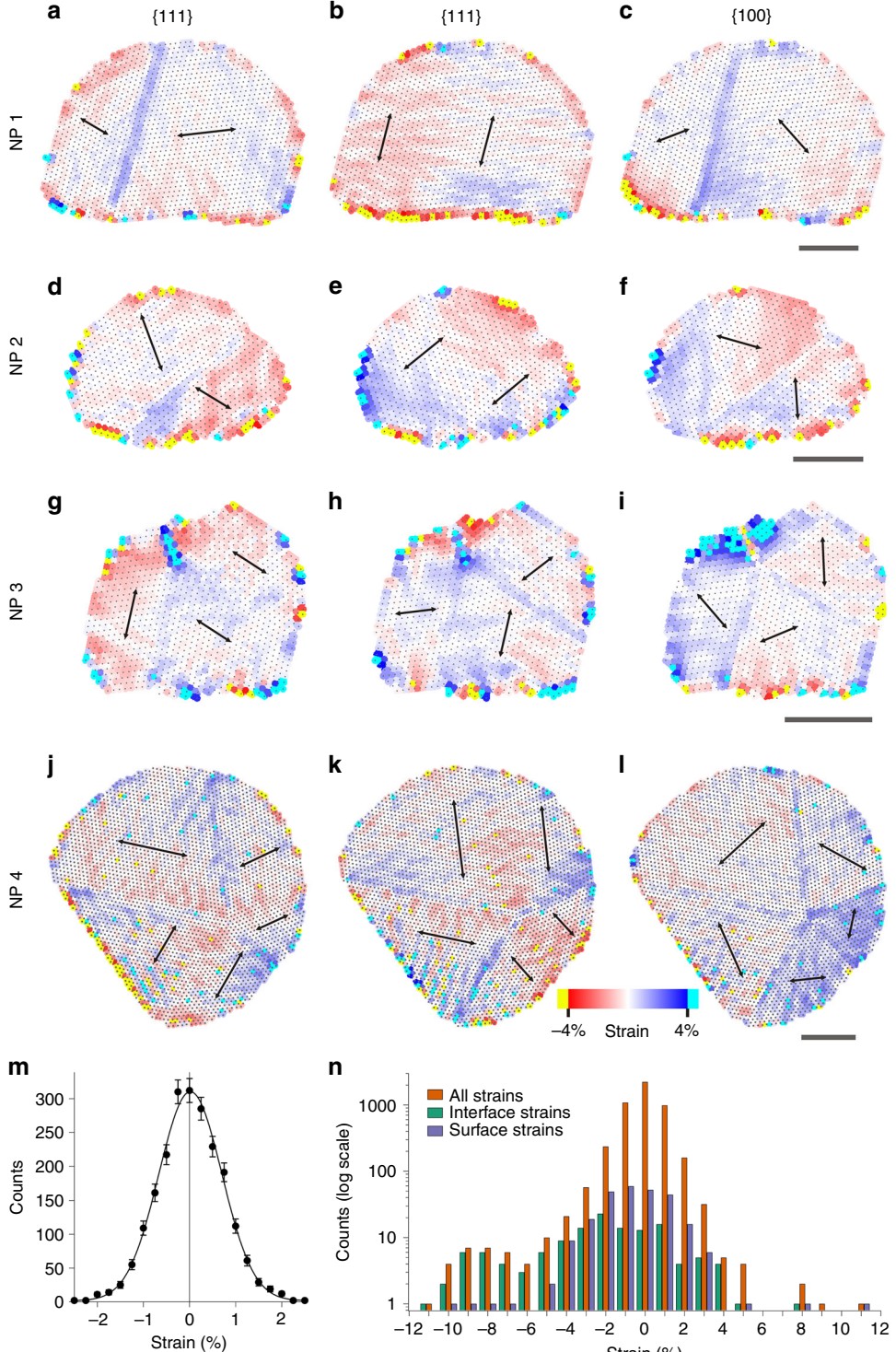

**Fig. 2** Atomic resolution strain behaviour of supported Pt NPs. **a–l** Atomic resolution strain maps within three crystallographic planes, as indicated above the NPs. The colour scale indicates variations in nearest-neighbour separations spanning from −4% (red, compressive) to 4% (blue, expansive). Strains above these thresholds are highlighted in yellow (compressive) and turquoise (expansive). Each scale bar corresponds to 3 nm. **m** Strain histogram from the unstrained reference area of NP 1, demonstrating <0.7% strain precision. The error bars correspond to the square root of the number of counts. **n** Strain histogram from the whole NP 1, vacuum surface and the NP–alumina interface

the intrinsic strains and are more frequently compressive, especially perpendicular to the interfaces. The interface between Pt and ceria displays matching periodic strain patterns between the Pt (Fig. 2j–l) and the ceria support (Supplementary Figure 5) that are associated with the interface dislocations.

The extrinsic interfacial strains observed here are likely caused by a combination of mechanisms, including thermal expansion coefficients mismatch[48,49], surface corrugation of the support[29,50] and lattice parameter mismatch[44,51]. Thermal expansion mismatch can cause residual strains when the catalyst is cooled from the

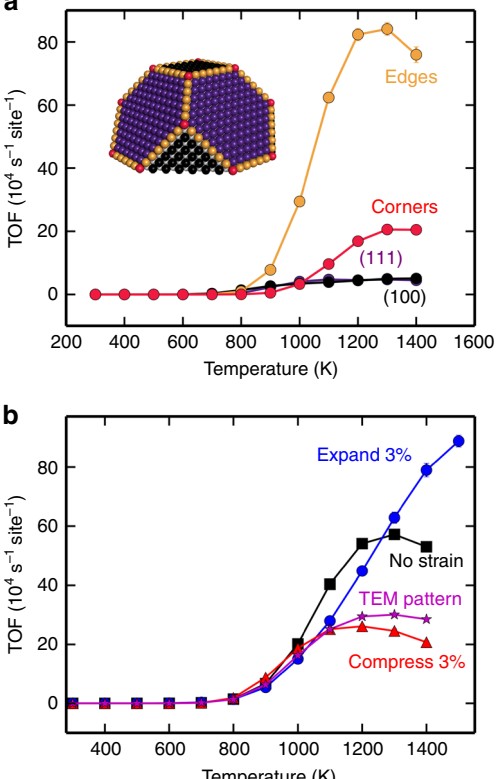

**Fig. 3** Light-off behaviour of CO oxidation determined by kinetic Monte Carlo simulations. **a** Light-off simulations showing site contributions to the total turnover frequency (TOF) for an unstrained NP. **b** Light-off simulations for various strain patterns. The simulated TOF is shown versus temperature. The data point markers are in most cases larger than the error bars. The error bars correspond to one standard deviation between 16 identically prepared simulations. Partial pressures: p(CO) = 20 mbar and p(O$_2$) = 10 mbar

calcination temperature at which the NPs are formed (≈500 °C) to room temperature. The two support materials used in this study have distinctly different morphologies. The ceria supports contain large single-crystalline particles with extended and atomically flat facets, while the γ-alumina supports consist of smaller irregularly shaped crystalline particles with no extended flat surface facets (Supplementary Figure 7). Despite these vastly different support morphologies, NPs on each exhibit similar strain magnitudes and a clear increase of strain at the NP–support interfaces compared to other regions of the NPs. The interaction mechanisms producing the interfacial strain should be active in most types of supported NP catalysts.

The projected strain measurements are quantitatively summarised in histograms, for example Fig. 2m, n for NP 1 (for other NPs see Supplementary Figure 4). The strain histogram from the unstrained reference area (Fig. 2m) confirms a <0.7% strain precision. Of the 4870 strain measurements from NP 1, 5.8% were larger than ±2%. The histograms in Fig. 2n compare the strain distribution from the whole NP to that of the surface and interfacial regions. This confirms that the large 5–10% compressive strains in this NP are primarily localised near the NP–support interface. Our high-precision strain measurements also reveal that strain is present in all investigated NPs, and likely in most of the NPs which are used in heterogeneous catalysis. This universal appearance motivates a further investigation into the impact of strain in NPs on their catalytic properties.

**Effects of strain on reaction kinetics**. To assess the influence of strain on nanoparticle catalysis, we theoretically investigated CO oxidation as a prototype reaction. We choose CO oxidation as this is a well-studied and technologically important reaction with dynamic reaction kinetics. The simulations were performed by a DFT-based scaling relation Monte Carlo method[38], where generalised coordination numbers are used as descriptors for adsorption energies and reaction barriers. As models for supported NPs, we consider 5.2 nm truncated octahedra cut in half. Various strain patterns were explored by assigning a strain to each site on the NP surface, and adjusting the reaction energies according to the scaling relations between reaction energies and strain (Supplementary Figure 9).

Figure 3a shows the temperature-dependent turnover frequency (TOF) demonstrating the light-off behaviour. The TOF is defined as the reaction rate per surface site. The reaction is explored at pressures relevant for a stoichiometric combustion engine, i.e., p(CO) = 20 mbar and p(O$_2$) = 10 mbar[52,53]. To uncover how different types of sites contribute to the total activity, the total TOF is separated into contributions from corner, edge, {100} and {111} sites. The activity is limited by CO poisoning at temperatures below 800 K and the poisoning is smallest on {111} sites which have the lowest CO adsorption energies. The activity is significant above 800 K, where the edges are most active, despite their low site fraction of 20%. Above 1000 K, corner sites also contribute to the overall activity. The edges are most active at high temperatures because they have a finite CO coverage, whereas the facets supply oxygen to the edge sites by diffusion[54], showing that long range kinetic coupling between edge and facet sites is crucial for the overall activity.

The light-off behaviour is investigated for three different strain patterns (Fig. 3b). The light-off temperature is commonly defined as the temperature where the TOF reaches half of its maximum. The simulations were performed for particles with 3% homogeneous expansion, 3% homogeneous compression and a strain pattern adopted from the STEM experiments. The strain pattern was obtained by random sampling of the experimental histogram in Fig. 2n, distinguishing between intrinsic and extrinsic strains. Irrespective of strain pattern, the activity is low below 800 K. At temperatures around 900 K, the NP under compressive strain yields the highest TOF, whereas the unstrained particle is most active in a temperature interval between 1000 and 1300 K. The NP under expansive strain has the highest activity at temperatures above 1300 K. The general behaviour is rationalised by noting that a compressive strain lowers the adsorption energies, which leads to a lower light-off temperature. Expansion increases the binding energies which prevents the reaction from becoming adsorption limited at high temperatures. The light-off temperature is lowest for the NP under compressive strain (944 K), higher for the experimentally obtained strain pattern (989 K) and the unstrained NP (1044 K) and highest for the NP under expansive strain (1167 K). This order is related to the fact that a lower CO binding energy reduces the effects of CO poisoning.

To quantify how different strain patterns affect the catalytic activity, we simulated eight qualitatively different strain patterns at a fixed temperature of 1100 K (Fig. 4). Patterns 0–2 are the unstrained, the 3% expanded and the 3% compressed NP, respectively. The strained NPs have TOFs that are ~35% lower than the unstrained NP. Pattern 3 is generated as a random sampling of the experimentally obtained histogram in Fig. 2n. To decouple the effects of extrinsic and intrinsic strain, Patterns 4 and 5 show the kinetics with strained interface and surface, respectively. The experimentally measured Pattern 3 lowers the TOF with respect to the unstrained NP by ~50%. The lowering is caused both by interface and surface strains which are not linearly additive owing to complex kinetic couplings[54] (Supplementary

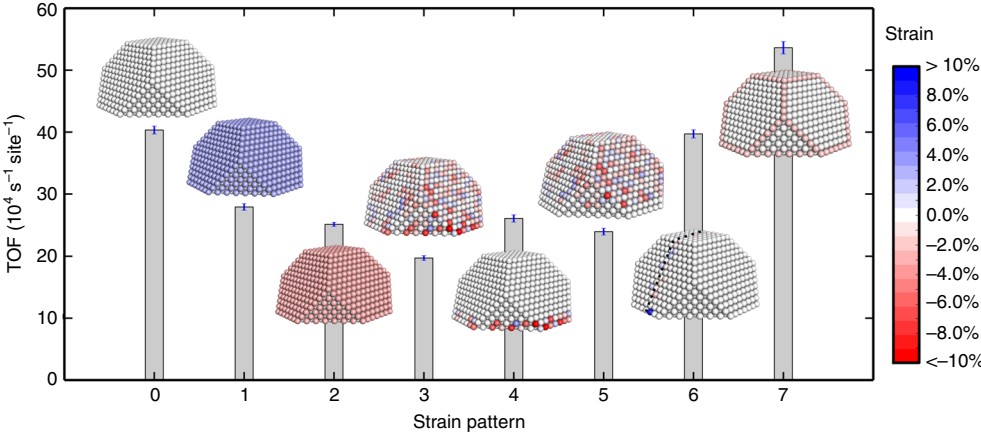

**Fig. 4** Predicting the influence of different strain patterns on the catalytic properties of a Pt NP. Turnover frequencies for strain patterns were simulated at 1100 K. Pattern 0: unstrained NP. Pattern 1: 3% expanded. Pattern 2: 3% compressed. Pattern 3: experimental distribution. Pattern 4: experimental strained interface. Pattern 5: experimental strained vacuum surface. Pattern 6: experimental strained twin boundary. Pattern 7: compressed edge and corner sites. The error bars correspond to one standard deviation between 16 identically prepared simulations. Partial pressures: p(CO) = 20 mbar and p(O$_2$) = 10 mbar

Note 3). Pattern 6 is generated from the measured twin boundary strain and has a minor effect on the activity, as only a few facet sites are strained. To mimic the experimentally observed preferential compression of edge and corner sites, we consider Pattern 7 where these sites are homogeneously compressed by 2%. This strain pattern increases the activity by ~30% with respect to the unstrained NP. The reason for the enhanced activity is the compression-induced lowering of the CO$_2$ formation barrier.

The activity of the different strain patterns can be understood by analysing how strain affects facets, edges and corners. In general, the highest activities are obtained when {111} and {100} facets are expanded, and the corners and edges are slightly compressed (for more details see Supplementary Note 3). It should be noted that the performance of a particular site depends non-linearly on the entire strain distribution. As various sites benefit from different values of strain, a selective strain distribution is generally needed to improve the catalytic performance of the NP. These simulations show that if the strain distribution can be controlled, it can be used to tune catalytic performance. Selection of different support materials, addition of surfactants or synthesis of shape- and size-selected NPs are examples of possible routes to control strain in NPs.

## Discussion

In this study, we measure site-specific strain in supported Pt NPs with atomic resolution and picometre precision, and reveal that all measured NPs are strained. Edge and corner surface sites, as well as regions close to twin boundaries, experience larger strain than surface facet sites. The largest magnitude and concentration of strains in the Pt NPs are observed at the γ-alumina and ceria-support interfaces. The presence of significant interfacial strain at both of these morphologically different support interfaces suggests that different strain generation mechanisms are active in the different systems and that the NP–support interface likely induces considerable strain in many types of supported nanocatalysts.

We present kinetic Monte Carlo simulations that predict the influence of strain on the catalytic activity of NPs using CO oxidation over Pt as a model reaction. By performing simulations of a complete NP model, we find that strain affects the light-off temperature and absolute turnover frequency. Different sites respond uniquely to strain, which is a consequence of the complex kinetic couplings present in NP catalysis. Future catalyst

design can benefit from the detailed knowledge obtained by the combination of high-precision experiments and advanced simulations, since it opens up possibilities to alter the catalytic activity by tuning the NP–support interaction and to separate structural from chemical effects at interfaces.

## Methods

**Pt nanocatalysts**. Supported platinum nanocatalysts were prepared by incipient-wetness impregnation[55]. An aqueous solution of tetraammineplatinum(II) nitrate ((NH$_3$)$_4$Pt(NO$_3$)$_2$, Alfa Aesar, 4 wt% Pt) was added to either γ-alumina or ceria nanopowder (Sigma-Aldrich, USA). Diluted ammonia solution (25% in water, Merck, Germany) was added to the mixture while stirring continuously to adjust the pH to 11. The resulting pastes were freeze-dried using liquid nitrogen, and the resulting dried powders were calcined in air at 500–540 °C for 1–3 h.

**STEM imaging experiments**. TEM specimens were prepared by crushing the catalyst powder in a mortar, dispersing it in alcohol and depositing it onto an ultra-thin carbon film supported on holey carbon (Ted Pella, USA). The specimens were cleaned using a plasma cleaner (Fischione Instruments (USA) model 1020, 25% oxygen in argon) for 20 s before the experiments to avoid contamination by carbon species.

STEM imaging was performed using an FEI (USA) Titan 80–300™ TEM/STEM instrument operated at acceleration voltages of 200 and 300 kV. The instrument is equipped with a field-emission electron source and a probe aberration corrector, resulting in sub-Ångström STEM spatial resolution. A reduced dose was achieved by using short-pixel dwell times (2–3 μs) and reducing the electron probe currents (~3 pA). STEM images were acquired using a HAADF detector, resulting in atomic number contrast that is well suited for imaging small metallic NPs[56]. A beam convergence semi-angle of 17.5 mrad and an inner collection semi-angle of 43 mrad were used.

High-precision STEM images were produced by acquiring an image series composed of tens to hundreds of fast exposure images of the same NP and performing post-acquisition image distortion correction and averaging[40]. All imaging parameters can be found in Supplementary Table 1. A pixelwise non-rigid registration (NRR) scheme[39] was used to correct the image distortions present in each frame of the image series. These distortions were caused by environmental vibrations, electronic instabilities and sample drift. After NRR, the low-distortion registered image series were averaged to enhance the SNR and ultimately the precision in locating atom column positions within the NPs.

**Strain analysis and visualisation**. The atomic column positions in a NP image were measured by fitting a two-dimensional Gaussian function to each atomic column[40]. In each grain of the NP a precision area was defined, which was separated by at least four atomic layers from any grain boundary, interface or surface. These precision areas were assumed to be mostly free of large strains and have crystal parameters close to bulk platinum. The x and y image pixel sizes were calibrated for each data set from the average nearest-neighbour distances in three crystallographic directions within the precision area of the largest grain. The image precision, defined as the standard deviation of the atomic column separations within the precision area of the largest grain[57,58], was measured to be between 1 and 4 pm in the data presented here.

We used two methods of visualising the projected NP lattice deformations: displacement maps and strain maps. Displacement maps were created using the following procedure: for each grain in the NP, an ideal periodic lattice was generated that represents the atomic column positions if no NP deformations were present. This ideal lattice was determined by creating a periodic lattice from the average interatomic separations in the precision area and registering it to the fit positions from the precision area. Then the periodic lattice was extended over all the atomic column positions in that grain. For the sites at twin boundaries, the average of the two ideal lattice positions from the two grains were used. In special cases when the ideal lattice positions at the twin boundaries deviated too much when considering different grains, they were not averaged and displacements from both ideal lattices are shown (for example Fig. 1f). The relative displacement of each atomic column fit position from the corresponding ideal lattice position was measured and indicated by an arrow in the displacement direction.

The displacement maps are an intuitive way of visualising the lattice deformation over whole NPs, partly because all the displacements can be shown within a single map. However, displacements accumulate and tend to be larger the further the atomic columns are away from the precision area since the ideal periodic lattice is aligned to the precision area. Therefore, nearest-neighbour strain maps are a more accurate and quantitative way to assess local lattice deformations.

Projected strain maps were created by comparing the measured nearest-neighbour distances in three crystallographic directions separately to reference values. Here, strain is defined as the deviation from the average atomic column spacing within the precision area divided by the average spacing. Each nearest-neighbour distance was treated individually and assigned a strain value. This implies that the same atomic column could be associated with compressive and expansive strains in different crystallographic directions. The local strain is indicated as a coloured region in the strain maps between each of the two neighbouring atomic columns. Separate strain maps were created for each of the three crystallographic directions. In order to observe trends in regions with small strains, the colour scale range was limited and strains with values outside the range of the colour scale were shown as yellow (compressive strain) or turquoise (expansive strain). Moreover, each strain that fell within the colour scale limits was averaged with the six nearest strain values in order to reduce noise. A small number of strains were measured to be larger than ±10%. However, these were associated with extremely thin atom columns at surface step edges and had displacements that were so large that the columns could not be attributed to real lattice sites. These sites were removed from the strain histograms.

**Kinetic Monte Carlo simulations**. NPs are characterised by a range of sites with different adsorption and reaction properties. It is challenging to describe such systems from first principles, in particular, as the number of unique atomistic configurations grows rapidly in the presence of adsorbate coverages. A pragmatic solution to the problem is to use scaling relations. It is established that Brønsted–Evans–Polanyi (BEP) relations can be used to obtain transition state energies from adsorption energies of reactant or products[59] and recently it was shown that adsorption energies on NP scale linearly with the generalised coordination number[60]. Thus, following Jørgensen et al.[38] we take advantage of such scaling relations to investigate the CO oxidation over Pt NPs. Strain is accounted for by the linear relation between adsorption energy and strain[32] (see SI).

The NPs were considered as truncated octahedra, which were cut in half to emulate a supported NP and the reaction energy landscape was established by use of DFT calculations, in particular the Vienna ab initio simulation package[61–64] with the revised Perdew–Burke–Ernzerhof[65] exchange-correlation functional. The interaction between the valence electrons and the core was treated within the projector-augmented wave scheme[66]. A kinetic cut-off of 450 eV was used to expand the Kohn–Sham orbitals.

Scaling relations of the adsorption energies as functions of the generalised coordination number were established using calculations on extended surfaces. The surfaces were modelled as a four layer slab and adsorbates were treated in $(2 \times 2)$ super-cells. A $(6 \times 6 \times 1)$ Monkhorst–Pack grid was used to sample the Brillouin zone.

A vacuum of 1.2 nm separated the periodic images in the direction perpendicular to the surface. Structural optimisations were performed using the BFGS line-search in the atomistic simulation environment[67] until all forces were below 0.05 eV Å$^{-1}$. The two bottom layers were fixed to the bulk positions during the relaxations. Vibrations were calculated using two-point differences with displacements of 0.01 Å. Gas-phase molecules were described in a $(30 \text{ Å} \times 30 \text{ Å} \times 30 \text{ Å})$ cubic cell with the proper spin states: singlet(CO, $CO_2$) and triplet($O_2$).

The energy barrier for $CO + O \rightarrow CO_2$ was calculated using climbing image nudged elastic band[68] with seven images, and initial interpolations with the image-dependent pair potential method[69]. Scaling relations of the adsorption energies as functions of the generalised coordination number were established using calculations on extended surfaces.

Moreover, a BEP relation was established between the transition state for $CO + O \rightarrow CO_2$ and the adsorption energies of $CO + O$[70]. Kinetic Monte Carlo[71] simulations were performed using the first reaction method with rate constants calculated by transition state theory[72]. The CO oxidation reaction was described

by:

$$CO(g) + * \leftrightarrow CO*$$
$$O_2(g) + 2* \leftrightarrow 2O* \quad (1)$$
$$CO* + O* \rightarrow CO_2(g) + 2*$$

Coarse graining of ontop, bridge and hollow positions was used to define one site. For each site, a strain was defined as a perturbation to the reaction energies. The reaction energies were modified by adsorbate–adsorbate interactions up to the first nearest neighbour. The barriers for CO diffusion were increased[73] to speed up convergence of the simulations. The convergence with respect to the kinetics was ensured prior to the final simulations. Further details of the simulations are described in Supplementary Notes 4–6.

**Data availability**. The raw STEM images of NP 1 are shown in Supplementary Movie 1. Raw STEM image series of NPs 1–4 are publicly available at https://osf.io/m8rbe. Additional data is available from the corresponding author upon request.

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

## Acknowledgements

This work has been performed within the Competence Centre for Catalysis, which is hosted by Chalmers University of Technology and financially supported by the Swedish Energy Agency and the member companies AB Volvo, ECAPS AB, Haldor Topsøe A/S, Scania CV AB, Volvo Car Corporation AB and Wärtsilä Finland Oy. The calculations were performed at C3SE (Gothenburg) and PDC (Stockholm) via a SNIC grant. The authors acknowledge the Knut and Alice Wallenberg Foundation, the Swedish Research Council (2016-05234), Chalmers Area of Advance Nanoscience and Nanotechnology, and the European Network for Electron Microscopy (ESTEEM2, European Union Seventh Framework Programme under Grant Agreement 312483-ESTEEM2 (Integrated Infrastructure Initiative-I3)) for financial support. We thank Laurence Marks and Magnus Skoglundh for fruitful discussions.

## Author contributions

All the authors conceived the study. T.N.P., E.O. and A.B.Y. designed the experiments. T.N.P. carried out the catalyst preparation and STEM experiments. M.J. and H.G. designed the simulations. M.J. wrote the SRMC code and performed the simulations. T.N.P. and A.B.Y. carried out the data analysis. A.B.Y. developed the data analysis software. E.O. and H.G. supervised the work. T.N.P. and M.J. co-wrote the manuscript. All the authors discussed the results and commented on the manuscript during its preparation.

## Additional information

**Competing interests:** The authors declare no competing interests.

