## [Peer Review File · Nature Communications]

Reviewers' comments:

Reviewer #1 (Remarks to the Author):

The manuscript presents interesting new experimental data from state of the art electron microscopy on strain distributions in supported metal nanoparticles and the relationship to the catalytic activity for the CO oxidation reaction is discussed. It is concluded a fraction of sites dominate the catalytic activity.

This is highly interesting since it shows for the first time the effect of local strain on catalytic activity. I suggest publication after the authors have considered some minor changes.

My suggestion is that the text, including the abstract and introduction could do with a slight rewording. It is my view that it overstates the generality and extent of the findings (which are interesting enough that this is not needed at all) somewhat. I am not convinced, for instance, that catalytic strain effects in nanoparticles are fundamentally different for strain effects on extended surfaces.

Reviewer #2 (Remarks to the Author):

This is an interesting and accomplished study of strain at the atomic level in platinum nanoparticles on metal oxide supports. Whilst atomic strain mapping of platinum on metal oxide supports has been performed previously, here the authors explore this in much more detail. Crucially, they explicitly link the strain to the catalytic activity of the nanoparticle. This relationship between strain and catalytic activity of nanoparticles is a concept that has been discussed in previous literature of metal oxide supported catalysts. However, this was largely speculative, and there are few conclusive references to back up these speculations. The authors of this study have delivered evidence for this strain-activity effect in a well-researched and thoroughly-implemented study. I believe that this work will be of enormous interest to researchers in heterogeneous catalysis, electrochemistry, and photochemistry. As such it should be published in Nature Comms after addressing the following important points.

Page 1:

In the first paragraph, "emission control" seems a little vague. Please be a little more specific and

descriptive (e.g. carbon emissions, pollutant emissions etc).

In the second paragraph: “modifies THE chemical properties”.

“there is a lack of atomically-resolved strain measurements of supported NPs.” - In references 27-31 there are several examples of exactly this. Whilst your study is clearly important, these previous works should be acknowledged more clearly and descriptively, whilst highlighting the novelty of your study compared to those.

Page 3:

How representative are the images? How many NPs were imaged? This could be made clearer in the main text.

“...which is expected by energetics” Please be more specific and descriptive about this statement.

“at typical” should be “under typical”

Perhaps highlight the concave re-entrant surface on the image with an arrow.

Figure 2: Are the first and second columns of Figure 2 both supposed to be $\{111\}$? If not, please fix this and check the rest of the manuscript carefully.

Page 5:

“we theoretically investigated CO oxidation as a prototype-reaction” - Is it expected that different reactions will have different effects? How far can this technique be generalized? What evidence do you have that it can be generalized? Why choose CO oxidation over other reactions? A little more discussion around this topic would be useful.

1st paragraph: NP -> NPs

“The reaction is explored at pressures relevant for a stoichiometric combustion engine, i.e. $p(\text{CO}) = 20 \text{ mbar}$ and $p(\text{O}_2) = 10 \text{ mbar}$ ” - References for these relevant conditions are required.

“These results indicate that long range kinetic coupling between edge and facet sites are crucial for the overall activity.” - Does this really follow from your results? It seems that the results you present only show that edges/corners are more active than facets. Reference [38] already implies that long-range kinetic coupling is crucial. Please clarify or modify your statement.

Figure 3a is described as “temperature dependent turnover frequency”, whereas 3b is described

in the context of “light-off behavior”. As far as I can tell, these graphs have the same axes and represent the same thing. Using two different contexts seems confusing. Can you discuss light-off behavior for 3a too? Please clarify these points.

“patterns, see Figure 3b” -> either “(see Figure 3b)”, or “as shown in Figure 3b”. Same for Figure 4.

“At temperatures around 900 K, the compressed particle yields the highest TOF” - Is this actually higher when the error is taken into account? The TOF is only very slightly higher in this region. Is it significant?

“Compressed particle” and “expanded particle” don’t feel quite right. Something like “Particle under compressive strain” may be more suitable.

“To decouple effects of extrinsic” -> “the effects”.

“The lowering is caused both by interface and surface strains which are not linearly additive owing to complex kinetic couplings” – Please explain this in more detail, and add a reference? This statement feels like a cop-out from providing a fuller explanation.

“This strain pattern increases the activity by ~30 % with respect to the unstrained NP. The reason for the enhanced activity is the compression induced lowering of the CO₂ formation barrier” - Are there any likely experimental synthesis routes to preferentially form this type of strain? E.g. the synthesis of nanocages, etc? Please speculate on the possibilities of actually creating such strained nanoparticles in order to guide experimentalists.

Page 6:

“These simulations show that if the strain distribution can be controlled, it can be used to tune catalytic properties.” – Can the strain distribution be controlled?? Is this actually feasible? How difficult/challenging will it be?

Whilst this study clearly shows that strain has an impact on the catalytic activity, it also shows that different strains in the same nanoparticle pull in different directions, and the overall effect in conventional NPs supported on metal oxide is negligible. I feel like your experiments show that a very specific type of strain is required for this to work in the real world. You should be clearer on the effect of the support on the strain and therefore the catalytic activity of the NPs. Is the metal oxide beneficial, detrimental, or unimportant? My own opinion is that this work shows that strain engineering is unlikely to be responsible for any improved activity previously observed in experimental systems, despite claims made to the contrary. Some further clarity and discussion on these matters should be included.

“Diluted ammonia solution” – Which supplier? What solvent?

We thank the Referees for appreciating our effort to investigate the role of strain in heterogeneous catalysis. We also thank them for the inspiring feedback and the valuable suggestions which guided us in improving the manuscript. A point-by-point response to the comments is given below.

Reviewer #1

The manuscript presents interesting new experimental data from state of the art electron microscopy on strain distributions in supported metal nanoparticles and the relationship to the catalytic activity for the CO oxidation reaction is discussed. It is concluded a fraction of sites dominate the catalytic activity. This is highly interesting since it shows for the first time the effect of local strain on catalytic activity. I suggest publication after the authors have considered some minor changes.

-- My suggestion is that the text, including the abstract and introduction could do with a slight rewording. It is my view that it overstates the generality and extend of the findings (which are interesting enough that this is not needed at all) somewhat. I am not convinced, for instance, that catalytic strain effects in nanoparticles are fundamentally different for strain effects on extended surfaces.

Response: We have changed the wording in the abstract and main text in order to not overstate the implications of our findings. Concerning the differences between extended surfaces and NPs, we see that they respond differently to strain as kinetic couplings are absent on extended surfaces. The kinetic couplings arise on NPs because of the presence of multiple types of sites. We studied these couplings in detail, see Ref. [54] (Jørgensen and Grönbeck, *Angew. Chem. Int. Ed.*, 57, p. 5086) of the revised manuscript.

The kinetic couplings are demonstrated in Supplementary Table S2 showing that the TOF of, e.g., edges depends both on the strain on the edges and on the facets. We agree with the Reviewer that these observations could have been stated more clearly. Thus, in the revised manuscript we added a remark on page 6, referring to Supplementary Note 3 and Supplementary Table S2 for further elaboration.

Reviewer #2

This is an interesting and accomplished study of strain at the atomic level in platinum nanoparticles on metal oxide supports. Whilst atomic strain mapping of platinum on metal oxide supports has been performed previously, here the authors explore this in much more detail. Crucially, they explicitly link the strain to the catalytic activity of the nanoparticle. This relationship between strain and catalytic activity of nanoparticles is a concept that has been discussed in previous literature of metal oxide supported catalysts. However, this was largely speculative, and there are few conclusive references to back up these speculations. The authors of this study have delivered evidence for this strain-activity effect in a well-researched and thoroughly-implemented study. I believe that this work will be of enormous interest to researchers in heterogeneous catalysis, electrochemistry, and photochemistry. As such it should be published in Nature Comms after addressing the following important points.

Page 1:

-- In the first paragraph, "emission control" seems a little vague. Please be a little more specific and descriptive (e.g. carbon emissions, pollutant emissions etc).

Response: The statement is more specific in the revised manuscript.

-- In the second paragraph: “modifies THE chemical properties”.

Response: This has been corrected.

-- “there is a lack of atomically-resolved strain measurements of supported NPs.” - In references 27-31 there are several examples of exactly this. Whilst your study is clearly important, these previous works should be acknowledged more clearly and descriptively, whilst highlighting the novelty of your study compared to those.

Response: The discussion of the previous literature has been expanded and clarified on page 1 of the revised manuscript.

Page 3:

-- How representative are the images? How many NPs were imaged? This could be made clearer in the main text.

Response: We agree with the referee that representativeness is a general and common concern when using TEM. We surveyed a large number of NPs (hundreds) and conclude that the ones presented in the manuscript are in size and morphology typical to the ones commonly found in the catalyst specimens. A major limiting factor for the number of NPs imaged with our high-precision technique was the complication of finding suitable NPs that could be tilted into a zone-axis orientation that enabled atomically-resolved STEM imaging. From all imaged NPs, we selected the ones with the highest image precision for the article, and these all show a common trend. Additional NPs were imaged, but the image precision was not good enough to reliably reveal the small-scale strain variations. The reduced precision can be caused by insufficiently precise alignment of the NPs with respect to the electron beam, specimen drift or contamination, or other material (for example oxide support) overlaying the NPs. In the future, as experimental procedures to determine NP orientations and precise tilting to zone-axis without exposure to high electron doses progress, additional studies with a higher yield of imaged NPs will be feasible. We added a comment about the representativeness of the included NPs with respect to size and morphology on page 2 in the revised manuscript.

-- “...which is expected by energetics” Please be more specific and descriptive about this statement.

Response: The statement has been clarified and a more suitable reference has been chosen.

-- “at typical” should be “under typical”

Response: This has been corrected.

-- Perhaps highlight the concave re-entrant surface on the image with an arrow.

Response: The concave re-entrant surface was marked with an arrow in Figure 1c.

-- Figure 2: Are the first and second columns of Figure 2 both supposed to be {111}? If not, please fix this and check the rest of the manuscript carefully.

Response: The two columns show indeed both strain in different {111}-type planes.

Page 5:

-- “we theoretically investigated CO oxidation as a prototype-reaction” - Is it expected that different reactions will have different effects?

Response: We expect that all reactions over nano-particles respond non-linearly to strain owing to the kinetic couplings, see Ref. [54] (Jørgensen and Grönbeck, *Angew. Chem. Int. Ed.*, 57, p. 5086). It has been demonstrated that adsorption energies on metals in general have similar strain dependences as CO and O, see e.g. Ref. [32] in the manuscript. Thus, CO oxidation is a reaction well suited for exploratory studies of the present kind.

-- How far can this technique be generalized? What evidence do you have that it can be generalized?

Response: The simulation technique can be generalized to all reactions on metal nanoparticles, where adsorption energies scale with the generalized coordination number. As the generalized coordination number is related to the d-band center [1-4], the method is applicable to a wide range of reactions.

[1] Hammer and Nørskov, *Nature* (1995), 376, p. 238

[2] Calle-Vallejo et al., *Nat. Chem.* (2015), 7, p. 403

[3] Calle-Vallejo et al., *Angew. Chem. Int. Ed.* (2014), 53, p. 8316

[4] Calle-Vallejo et al., *Science* (2015), 350, p. 185

-- Why choose CO oxidation over other reactions? A little more discussion around this topic would be useful.

Response: We choose CO oxidation as it is a well-studied reaction with a dynamic reaction kinetics. In addition, it is a technologically very important reaction taking place in all three-way-catalysts in gasoline cars. We have added a remark on page 5 to better motivate the choice of CO oxidation as the model reaction.

-- 1st paragraph: NP -> NPs

Response: This has been corrected.

-- "The reaction is explored at pressures relevant for a stoichiometric combustion engine, i.e. $p(\text{CO}) = 20 \text{ mbar}$ and $p(\text{O}_2) = 10 \text{ mbar}$ " - References for these relevant conditions are required.

Response: We have included references to show that the used pressures are in the same order of magnitude as from combustion engines.

-- "These results indicate that long range kinetic coupling between edge and facet sites are crucial for the overall activity." - Does this really follow from your results? It seems that the results you present only show that edges/corners are more active than facets. Reference [38] already implies that long-range kinetic coupling is crucial. Please clarify or modify your statement.

Response: We agree with the Reviewer that the character of the kinetic couplings do not follow from the discussion of Figure 3 and 4 in the original manuscript. We have in the revised manuscript added a reference and rephrased the wording on page 5. The couplings are presented in Supplementary Note 3. The figure below demonstrates the couplings by disabling the (111) facets from the reaction. The figure shows the TOF of CO and O reacting from different site combinations. Disabling adsorption and reaction on (111) clearly affects the TOFs of edges and corners.

Figure 1: Turnover frequency of a 5.2 nm unstrained particle with all sites active (purple) and all (111) facets disabled (orange). Temperature: 1100 K, pressures CO: 20 mbar, O₂: 10 mbar.

-- Figure 3a is described as “temperature dependent turnover frequency”, whereas 3b is described in the context of “light-off behavior”. As far as I can tell, these graphs have the same axes and represent the same thing. Using two different contexts seems confusing. Can you discuss light-off behavior for 3a too? Please clarify these points.

Response: We thank the reviewer for pointing out this possible confusion. Both figure captions use “light-off behaviour” in the revised version. Moreover, in the revised manuscript we have framed the presentation of Figure 3a to discuss the light-off behavior.

-- “patterns, see Figure 3b” -> either “(see Figure 3b)”, or “as shown in Figure 3b”. Same for Figure 4.

Response: This has been corrected.

-- “At temperatures around 900 K, the compressed particle yields the highest TOF” - Is this actually higher when the error is taken into account? The TOF is only very slightly higher in this region. Is it significant?

Response: We agree that it is difficult to see the error-bars in Figure 3. The figure below shows that the error-bars and data points do not overlap. In the revised manuscript we have added a sentence to the figure caption stating that the markers are in most cases larger than the error-bars. The data used in the figure is:

TOFS (1/(site*s)) at 900K (Unstrained, expanded 3%, compressed 3%, experimental pattern):
66505, 54014, 87560, 63015

Error-bars (1/(site*s)) at 900K (Unstrained, expanded 3%, compressed 3%, experimental pattern) :
1798, 863, 1650, 1801

-- *“Compressed particle” and “expanded particle” don’t feel quite right. Something like “Particle under compressive strain” may be more suitable.
“To decouple effects of extrinsic” -> “the effects”.*

Response: This has been edited where suitable.

-- *“The lowering is caused both by interface and surface strains which are not linearly additive owing to complex kinetic couplings” – Please explain this in more detail, and add a reference? This statement feels like a cop-out from providing a fuller explanation.*

Response: We agree that it was difficult to follow and have added a reference to our recent paper ([54]) as well as a reference to Supplementary Note 3.

-- *“This strain pattern increases the activity by ~30 % with respect to the unstrained NP. The reason for the enhanced activity is the compression induced lowering of the CO₂ formation barrier” - Are there any likely experimental synthesis routes to preferentially form this type of strain? E.g. the synthesis of nanocages, etc? Please speculate on the possibilities of actually creating such strained nanoparticles in order to guide experimentalists.*

Page 6:

-- *“These simulations show that if the strain distribution can be controlled, it can be used to tune catalytic properties.” – Can the strain distribution be controlled?? Is this actually feasible? How difficult/challenging will it be?*

Response (combined for the two comments): Since the aim of our work is to advance the understanding of the impact of strain on catalytic activity and not to develop recipes for strain engineering, we have chosen not to discuss possibilities of strain engineering in detail. Engineering NP strain is complex and could be difficult to achieve at industrial scale. We believe it’s not impossible though, and future studies are required to explore different approaches. The referee’s comments regarding speculation about possible routes of manipulating strain are good, and this will help guide experimentalists in future studies. Therefore we added a short remark on this topic in the last paragraph of our results / discussion section on page 6 of the revised manuscript.

-- *Whilst this study clearly shows that strain has an impact on the catalytic activity, it also shows that different strains in the same nanoparticle pull in different directions, and the overall effect in conventional NPs supported on metal oxide is negligible. I feel like your experiments show that a very specific type of strain is required for this to work in the real world. You should be clearer on the effect of the support on the strain and therefore the catalytic activity of the NPs. Is the metal oxide beneficial, detrimental, or unimportant? My own opinion is that this work shows that strain engineering is unlikely to be responsible for any improved activity previously observed in experimental systems, despite claims made to the contrary. Some further clarity and discussion on these matters should be included.*

Response: We would like to point out that strained NPs like the ones observed in our study appear to be the standard, not the exception. This is in contrast to the mindset that NPs are inherently unstrained and can be strained to tune their properties. We added a statement about this to pages 5 and 6 of the revised manuscript. Due to the universality of the phenomenon, we established

methods to investigate NP strain and its impact on the catalytic properties. We demonstrate that strain of the magnitude which we experimentally observe has an impact on the catalytic activity, and we agree with the referee that quite specific strain might be needed to purposefully enhance catalytic properties. We also think that further studies are needed to answer if and how strain engineering can be successfully used for activity enhancement. Strain is only one of many factors influencing catalytic properties, and is likely not the dominant one. We still feel it is important to understand the consequences of strain, and expect that future studies will illuminate if and how strain modification can be utilised.

-- *“Diluted ammonia solution” – Which supplier? What solvent?*

Response: Information about the supplier (Merck, Germany), concentration and solvent (25% in water) has been added to the revised manuscript.

Reviewers' Comments:

Reviewer #2 (Remarks to the Author):

I am convinced that the author has dealt with all of the reviewer comments clearly and fully. Therefore I recommend publishing this manuscript.